# Medical use of cocaine and perioperative morbidity following sinonasal surgery—A population study

**S. Danielle MacNeil**[1]*, **Brian Rotenberg**[1], **Leigh Sowerby**[1], **Britney Allen**[2],
**Lucie Richard**[2], **Salimah Z. Shariff**[2]

**1** Department of Otolaryngology-Head and Neck Surgery, Schulich School of Medicine and Dentistry,
London, Ontario, Canada, **2** ICES Western, London, Ontario, Canada

* Danielle.macneil@lhsc.on.ca

## Abstract

### Background

Topical cocaine is favoured by many surgeons for sinonasal surgery due to its superior
vasoconstrictive and anesthetic properties. However, historical reports suggesting cocaine
is associated with an increased risk of cardiac events have led many surgeons to turn to
alternative topical medications. The objective of this study was to determine whether
cocaine use during sinonasal surgery is associated with an increased risk of perioperative
cardiac events and death.

### Methods

We conducted a population-based analysis of patients undergoing sinonasal surgery from
2009–2016 using linked administrative health care data sets in Ontario, Canada. We com-
pared patients treated at institutions that primarily use topical cocaine (exposed group) to
those treated at institutions that do not use cocaine (unexposed group). Our primary out-
come was a composite of major cardiac events or all-cause mortality within 48 hours of sur-
gery. Due to low event rates, the outcome was compared using a Fisher's exact test.

### Results

Of 10,549 patients who were included in the study, 27.4% were treated at an institution that
uses topical cocaine. The rate of the composite of perioperative major cardiac event or all-
cause mortality within 48 hours of surgery in the exposed and unexposed groups was,
$\leq$0.2% and 0 (p-value>0.05), respectively.

### Conclusions

In this large real-world cohort of patients undergoing sinonasal surgery, there does not
appear to be any significant increased risk of morbidity or mortality associated with cocaine
use. These findings have important implications for surgeons performing this procedure.

pone.0236356

Medicine and Public Health, AUSTRALIA

**Data Availability Statement:** The dataset used in
this study is held securely in coded format at ICES.
ICES is a prescribed entity under section 45 of
Ontario's Personal Health Information Protection

Act. Section 45 authorizes ICES to collect personal health information, without consent, for the purpose of analysis or compiling statistical information with respect to the management of, evaluation or monitoring of, the allocation of resources to or planning for all or part of the health system. Legal restrictions and data sharing agreements prohibit ICES from making the dataset publicly available. Access may be granted to those who meet the conditions for confidential access, available at https://www.ices.on.ca/DAS. SS holds an appointment as an ICES Scientist, which enabled access to ICES data. Data access is available to external public sector researchers either through collaboration with an ICES scientist or directly, following project approval, via a secure online desktop infrastructure (see above link for details).

**Funding:** This study was supported by ICES, which is funded by an annual grant from the Ontario Ministry of Health and Long-Term Care (MOHLTC). The study was completed at the ICES Western site, where core funding is provided by the Academic Medical Organization of Southwestern Ontario, the Schulich School of Medicine and Dentistry, Western University, and the Lawson Health Research Institute. This study also received funding from the St. Joseph's Health Care Foundation grant no. 012-1718 to BR. Parts of this material are based on data and information compiled and provided by CIHI. The analyses, conclusions, opinions and statements expressed herein are solely those of the authors and do not reflect those of the funding or data sources; no endorsement is intended or should be inferred.

**Competing interests:** The authors have declared that no competing interests exist.

## Introduction

Cocaine has been widely used in all forms of nasal surgery, in particular septoplasty, rhinoplasty, and endoscopic sinus surgery [1]. In the United States, approximately 600,000 ambulatory sinonasal surgeries are performed each year [2]. Cocaine has been favored for decades as the optimal agent for its long-lasting local vasoconstriction and profound sensory nerve inhibition properties [1]. For nasal and sinus surgery in particular, topical vasoconstriction is essential to minimize bleeding for adequate visualization of anatomic landmarks and to prevent intraoperative hemorrhage [3]. Recently, the safety of cocaine has been questioned, citing historical data that suggested an increased risk of perioperative cardiac morbidity and mortality [4]. These concerns are largely based on low quality of evidence; however the resulting concern regarding patient safety and medicolegal uncertainty has led many surgeons to avoid using cocaine [1, 5, 6].

The safety of cocaine has been questioned due to an apparent increased risk of arrhythmia, hypertension and serious adverse cardiac events including death [1, 6]. The literature describing adverse events in patients receiving intranasal cocaine have several limitations [4]. The report of perioperative cardiac events in patients receiving intranasal cocaine are largely case reports and case series [4]. Further, the majority of reports are in patients receiving doses of cocaine far larger and in higher concentrations than current practice [4]. A small randomized trial of 37 patients has demonstrated no adverse events in patients receiving either cocaine or an alternative [7]. However, this trial was not powered to detect a difference in perioperative morbidity or mortality. To date, no adequately powered clinical trial or observational study has demonstrated a convincing association between the use of cocaine and adverse perioperative events. This represents a major gap in medical knowledge that directly relates to perioperative patient outcomes.

To definitively determine whether cocaine use in sinonasal surgery is associated with an increased risk of perioperative morbidity and mortality, we chose to utilize the population databases in Ontario, to compare perioperative cardiac events and death in patients undergoing sinonasal surgery at institutions that use cocaine versus institutions that use alternative topical medications. We hypothesized that the intraoperative use of medicinal grade cocaine in patients without a history of cardiac disease would not infer an increased risk during sinonasal surgery.

## Methods

### Design and setting

The province of Ontario, Canada has a population of over 13 million people. The residents of Ontario have universal access to hospital care and physician services. Each encounter with the healthcare system is recorded in large, population-based, linked health care databases that are held at ICES (formerly referred to as the Institute of Clinical Evaluative Sciences). We performed a population-based retrospective cohort study of all patients who underwent sinus surgery or had a septoplasty between April 1, 2009 and March 31, 2016. Guidelines for observational studies as outlined in the STROBE guidelines were followed for this study [8].

### Data sources

The following linked administrative databases at ICES were used: the Canadian institute for Health information's discharge abstract database (CIHI-DAD) which records all admission to hospitals and includes information on diagnoses and procedures performed [9]; the Ontario Health Insurance Plan database (OHIP) contains information on all fee-for-service physician

claims for inpatient and outpatient services [10]; the Registered Persons Database (RPDB) which contains vital statistics on all permanent residents of Ontario [11]; the National Ambulatory Care Reporting System (NACRS) database which collects data on all ambulatory care visits, including day surgery, outpatients' clinics, cancer clinics, and emergency department visits; the ICES-derived Ontario Diabetes Database (ODD) [12]; the ICES derived the Congestive Heart Failure (CHF) [13] database; the ICES-derived Hypertension database (HYPER) [14]; and the ICES-derived Ontario Myocardial Infarction Database (OMID) [15]; and the OHIP database to identify patients who had sinonasal surgery. To define patient characteristics, baseline comorbidities and patient outcomes a combination of CIHI-DAD, OHIP, ODD, CHF, HYPER, OMID, NACRS and RPDB databases were used. Diagnoses and procedures were defined using the international Classification of Diseases, ninth revision (ICD-9; pre-2002), 10th revision (ICD-10; post- 2002), and Canadian Classification of Health Interventions and Canadian Classification of Diagnostic, Therapeutic and Surgical Procedures codes. These data holdings were linked using unique encoded identifiers and analyzed at ICES.

## Participants

Patients 18 years of age and older with a billing code for sinonasal surgery between the years of 2009 to 2016 were included. Patients were identified if there was a physician services billing code for polypectomy, ethmoidectomy, or septoplasty (Z304, Z305, M083, M012). We excluded pediatric patients and those with invalid ages (<18 or >105), non-Ontario residents and those who were not treated at one of the six candidate institutions (see Exposure ascertainment below). We further restricted our cohort to patients with no prior history of myocardial infarction, percutaneous coronary intervention or coronary artery bypass grafting within 5 years of surgery, and no prior history of congenital heart disease within 10 years of the sinonasal surgery. Patients who underwent more than one sinonasal surgery during the accrual period were restricted to their first surgery. The date of the procedure code for sinonasal surgery served as the start time for follow-up (also referred to as the index date). We obtained information on the patient's baseline characteristics (age, sex, socioeconomic status, residency status) on the surgery date. We also obtained information on the provider (years in practice, number of cases performed per year). We assessed the comorbidity status of our cohort using the health care records in the 3 years preceding the surgery date using Resource Utilization Bands (RUBs) defined by the Johns Hopkins Adjusted Clinical Group (ACG) Classification System [16]. The ACG system helps to describe the past and the future of health care utilization and costs [16]. RUBs are a marker of resources utilization, where 0 corresponds to nonusers and 5 designates patients with very high levels of morbidity and resources utilization.

## Exposure ascertainment

As cocaine use is not coded in administrative databases, exposure was ascertained using institution at which the surgery was performed. We determined, through personal communication, that two institutions in Ontario use cocaine routinely (>95% of the time) for sinonasal surgery. Administration of topical anesthetic in the cocaine group was 1.4% cocaine in 1:10,000 epinephrine solution. Additionally, four institutions in Ontario had no access to cocaine during the study period; therefore it was not possible for patients who had surgery at these institutions to have been treated with cocaine. The topical anesthetics used at institutions that did not use cocaine included various agents: 1:10,000 epinephrine; 1:100,000 epinephrine; 0.05% oxymetazoline; or 1–2% lidocaine.

## Outcome measures

Our primary outcome measure was a composite of a major cardiac event—including myocardial infarction, cardiac procedure including cardiac artery bypass grafting, percutaneous coronary intervention or death—within 48 hours of surgery. Our secondary outcome extended the window of the primary outcome to 30 days.

## Statistical analysis

Baseline characteristics were compared using standardized differences, which measures the difference in the mean of a variable between two groups divided by an estimate of the standard deviation of that variable among both groups [17]. A standardized difference >0.1 is considered an important difference [17]. Differences in rates of cardiac events between exposure groups were compared using multiple logistic regression or Fisher's exact test [18], as appropriate based on the number of events. All statistical analyses were conducted with SAS version 9.4 (SAS Institute, Cary, NC).

## Ethics approval

ICES is a designated prescribed entity under Section 45 of the Personal Health Information Protection Act (PHIPA). Participant informed consent was not required for this study. All data was anonymized before it was accessed. The study was approved by the research ethics board of Sunnybrook Health Sciences Centre.

## Patient involvement

No patients were involved in setting the research question or the outcome measures, nor were they involved in developing plans for implementation of the study. No patients were asked to advise on interpretation or analysis of results. There are no plans to disseminate the results of the research to study participants or the relevant patient community.

## Results

Cohort selection is presented in Fig 1. After restricting to the six candidate institutions, there were 10,549 surgeries remaining, with 2,887 (27.4%) performed at an institution that uses topical cocaine for intraoperative hemostasis and anesthesia.

Differences were observed between patient groups. Patients who were treated at a cocaine-using institution were more likely to be older, have hypertension, and have been operated on by a surgeon with fewer years in practice, who performed fewer endoscopic sinus surgeries per year (Table 1).

Rates of primary and secondary outcomes are presented in Table 2. The rate of the primary composite outcome of perioperative major cardiac event or all-cause mortality within 48 hours in the exposed and unexposed groups was, ≤0.2% and 0 (p-value>0.05), respectively. The rate of the secondary outcome of perioperative major cardiac event or all-cause mortality within 30 days in the exposed and unexposed groups was, 0.24 and 0.08 (p-value 0.056), respectively.

Due to the low event rates observed, Fisher's exact test was applied to each of the primary and secondary outcomes. Due to potential risk of patient re-identification, institutional policies prohibit the presentation of results of 5 or fewer individuals. Furthermore, exact p-value for the primary outcome could not be presented to avoid the exact number of event rates from being back calculated.

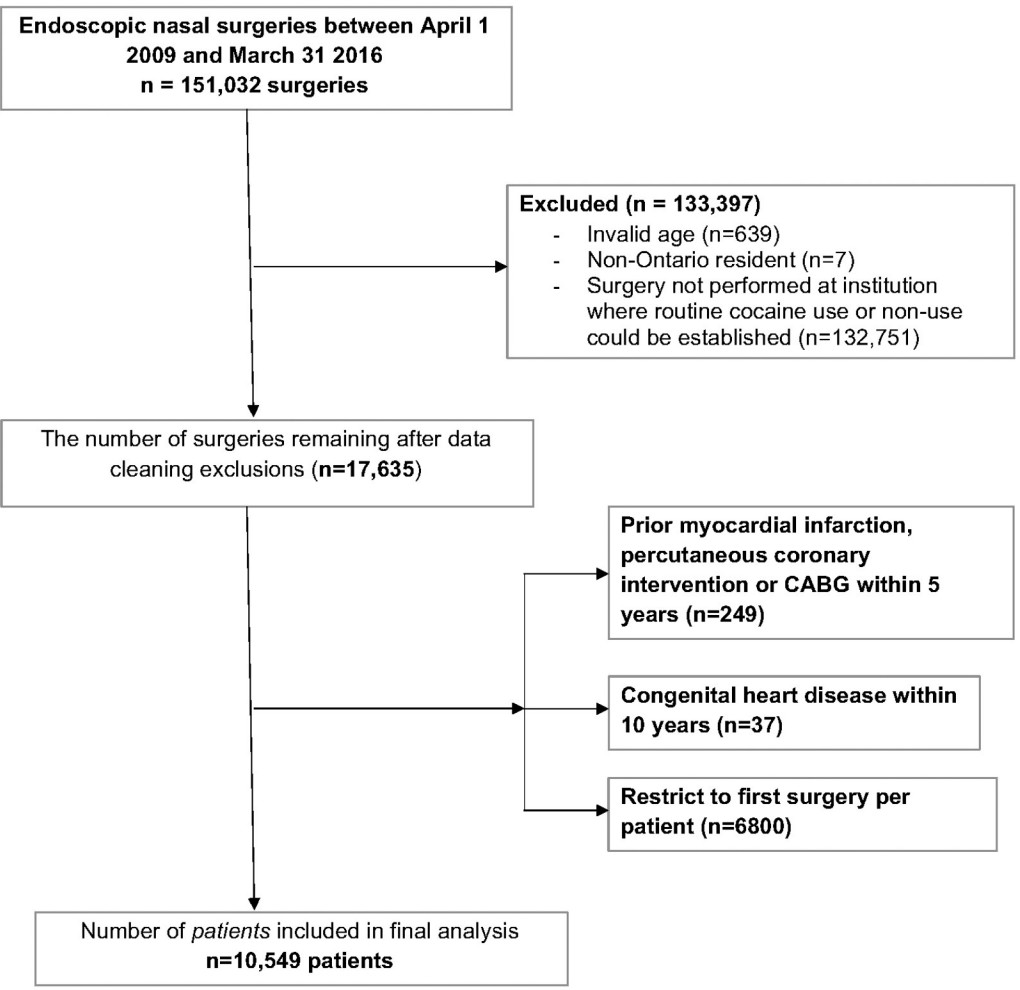

**Fig 1. Inclusion/Exclusion flow chart.**

## Discussion

### Summary of the main results

In this large population-based study, we investigated cardiac events after sinonasal surgery comparing institutions that use cocaine versus those that do not. Overall, we found a very low rate of cardiac events and mortality. There was no statistically significant event rate difference (major cardiac event and death) in the patients who were treated at institutions that used cocaine versus those that did not. Due to the low event rate we were not able to perform an adjusted analysis which resulted in some baseline differences between the exposed and unexposed groups.

We report here, the first cohort study of topical cocaine versus other topical anesthetics for use in sinonasal surgery examining cardiac outcomes and death. In spite of the low event rate we observed, this is the best available data demonstrating no difference in cardiac events and mortality in patients that received intranasal cocaine versus those that didn't given the large sample size of our study. Due to the single payer healthcare system in Ontario, we were able to comprehensively capture patients undergoing the selected surgical procedures. We were confident through communication with the surgeons performing sinonasal surgery at the

**Table 1. Patient characteristics, by exposure to topical cocaine during surgery.**

| | Total (N = 10,549) | Cocaine (N = 2,887) | Not cocaine (N = 7,662) | Standardized Difference |
|---|---|---|---|---|
| **Demographics** | | | | |
| Age at Index Date | | | | |
| Mean (SD) | 47.04 ± 15.78 | 49.24 ± 16.41 | 46.22 ± 15.46 | 0.19 |
| Median (IQR) | 47 (35–58) | 50 (37–61) | 46 (34–57) | |
| Female, N (%) | 4,432 (42.0%) | 1,251 (43.3%) | 3,181 (41.5%) | 0.04 |
| Income quintile, N (%) | | | | |
| Quintile 1 (lowest) | 1,688 (16.0%) | 419 (14.5%) | 1,269 (16.6%) | 0.06 |
| Quintile 2 | 1,951 (18.5%) | 560 (19.4%) | 1,391 (18.2%) | 0.03 |
| Quintile 3 | 2,017 (19.1%) | 559 (19.4%) | 1,458 (19.0%) | 0.01 |
| Quintile 4 | 2,269 (21.5%) | 654 (22.7%) | 1,615 (21.1%) | 0.04 |
| Quintile 5 (highest) | 2,568 (24.3%) | 681 (23.6%) | 1,887 (24.6%) | 0.02 |
| Missing[a] | 56 (0.5%) | 14 (0.5%) | 42 (0.5%) | 0.01 |
| Rural, Yes, N (%) | 1,144 (10.8%) | 453 (15.7%) | 691 (9.0%) | 0.2 |
| Year of Cohort Entry, N (%) | | | | |
| 2009 | 1,176 (11.1%) | 352 (12.2%) | 824 (10.8%) | 0.05 |
| 2010 | 1,094 (10.4%) | 363 (12.6%) | 731 (9.5%) | 0.1 |
| 2011 | 1,314 (12.5%) | 353 (12.2%) | 961 (12.5%) | 0.01 |
| 2012 | 1,383 (13.1%) | 348 (12.1%) | 1,035 (13.5%) | 0.04 |
| 2013 | 1,462 (13.9%) | 375 (13.0%) | 1,087 (14.2%) | 0.03 |
| 2014 | 1,439 (13.6%) | 350 (12.1%) | 1,089 (14.2%) | 0.06 |
| 2015 | 1,344 (12.7%) | 348 (12.1%) | 996 (13.0%) | 0.03 |
| 2016 | 1,337 (12.7%) | 398 (13.8%) | 939 (12.3%) | 0.05 |
| **Comorbidities in the previous 5 years** | | | | |
| Congestive heart failure, N (%) | 133 (1.3%) | 46 (1.6%) | 87 (1.1%) | 0.04 |
| Diabetes, N (%) | 1,042 (9.9%) | 306 (10.6%) | 736 (9.6%) | 0.03 |
| Hypertension, N (%) | 2,729 (25.9%) | 875 (30.3%) | 1,854 (24.2%) | 0.14 |
| Resource Utilization Band [16], N(%) | | | | |
| 0–2 | 641 (6.1%) | 216 (7.5%) | 425 (5.5%) | 0.08 |
| ≥3 | 9,908 (93.9%) | 2,671 (92.5%) | 7,237 (94.5%) | |
| **Surgeon Characteristics at Index Date** | | | | |
| Years in Practice | | | | |
| Mean (SD) | 19.58 ± 12.42 | 15.08 ± 10.11 | 21.25 ± 12.78 | 0.53 |
| Median (IQR) | 14 (10–26) | 11 (9–15) | 18 (10–28) | |
| Relative volume of endoscopic sinus surgeries in prior year, N (%) | | | | |
| Quintile 1 (lowest) | 2,154 (20.4%) | 462 (16.0%) | 1,692 (22.1%) | 0.16 |
| Quintile 2 | 2,197 (20.8%) | 367 (12.7%) | 1,830 (23.9%) | 0.29 |
| Quintile 3 | 1,880 (17.8%) | 312 (10.8%) | 1,568 (20.5%) | 0.27 |
| Quintile 4 | 2,274 (21.6%) | 370 (12.8%) | 1,904 (24.8%) | 0.31 |
| Quintile 5 (highest) | 2,044 (19.4%) | 1,376 (47.7%) | 668 (8.7%) | 0.96 |

institutions selected that they either primarily used cocaine for topical treatment of the nasal cavity or their institution had a policy of not using cocaine. Further, we were able to accurately capture hospital readmission and complications occurring anywhere in the province, including institutions other than where the index surgery was performed, due to the reporting of these events within the healthcare system in Ontario.

We observed baseline differences in the groups which received cocaine versus those that did not receive cocaine. Patients in the cocaine group were more likely to be older, live in a

**Table 2. Descriptive statistics of outcomes, by exposure to cocaine during surgery.**

| | Exposed to cocaine (N = 2,887) | | Unexposed to cocaine (N = 7,662) | | P Value[b] |
|---|---|---|---|---|---|
| | N of events | Event rate (%) | N of events | Event rate (%) | |
| Major cardiac event or death within 48 hours of surgery, N (%) | ≤5[a] | ≤0.2[a] | 0 | 0 | >0.05[a] |
| Major cardiac event or death within 30 days of surgery, N (%) | 7 | 0.24 | 6 | 0.08 | 0.056 |

[a] In accordance with ICES privacy policies, cell sizes less than or equal to five cannot be reported.

[b] Differences in rates of cardiac event or death between exposure groups were compared using Fisher's exact test.

rural location and have hypertension. The province from which were drew our sample is heterogenous, containing densely populated areas receiving referrals primarily from within cities, as well as less populated smaller cities with referral patterns from primarily rural locations. As we determined our treatment (cocaine) and control (not cocaine) groups by centres that use cocaine versus those that don't, the differences are likely related to the surrounding geography. Patients who are older are more likely to live in rural locations [19] and to have hypertension [20].

We also observed baseline differences between groups with respect to surgeon years in practice and volume of surgeries per year, with surgeons in the control (not cocaine) group having higher number of years in practice. While surgeons with fewer years in practice usually have less access to operating room time and therefore would perform fewer surgeries per year, the difference found likely reflects the small number of institutions that were represented in our study, with most surgeries performed by a small number of surgeons. Interestingly, and contrary to our findings, two survey studies performed in Canada and in the United States demonstrated that surgeons who had fewer years in practice were less likely to use intranasal cocaine [1, 6]. Although these survey studies indicate that younger surgeons are less likely to use cocaine, the decision about whether to use cocaine for intranasal surgery is more likely driven by fear of medicolegal concern if an adverse event occurs, the availability of alternate intranasal medications and institutional policies that prohibit intraoperative use of cocaine due to concerns with the storage and dispensing of this controlled substance.

Our results are consistent with previous large studies in the literature indicating a low rate of cardiac events and mortality following sinonasal surgery. Bhattacharyya (2010) studied perioperative outcomes in over 600,000 patients undergoing sinonasal surgery in the United States over a one year period and found that there were no cases of cardiac arrest [2]. The lower rate of cardiac arrest observed in this study may be explained by the use of the National Survey of Ambulatory Surgery database which includes both hospital–based ambulatory surgery and freestanding ambulatory surgery centers [2]. All of the institutions included in our study were tertiary care academic hospitals and as a result patients have more comorbidities and the surgeries are more complex requiring in some cases a fellowship-trained Rhinologist.

Several trials have been conducted on patients undergoing sinonasal surgery using topical cocaine compared to other agents. The primary outcomes of these studies include pain perception [21, 22], plasma absorption of intranasal cocaine [23–25], surgical field visualization and intraoperative bleeding [7, 26–28], and adverse events (including cardiovascular changes, electrocardiogram changes, cardiovascular events and mortality) [2, 25–30]. There was no difference in intraoperative bleeding or surgical field visualization amongst the studies reported in the literature [7, 26–28]. Further, none of the trials in the literature demonstrate an increased risk of cardiovascular morbidity or mortality associated with the use of intranasal cocaine which is in keeping with the results of our study [2, 25–30]. The reports of adverse cardiovascular events associated with the use of intranasal cocaine are from case studies [4]. Our study

along with the trials reported in the literature demonstrate that there is no increased risk of cardiovascular related morbidity or morality associated with the use of cocaine for sinonasal surgery.

Our study carries some limitations, primarily related to the observational design. Unmeasured (and unmeasurable) residual confounding could not be accounted for. Due to the low event rate we were unable to perform an adjusted analysis comparing our two groups. Difference in age, comorbidity and surgeon experience might have contributed to our findings. We were not able to obtain details on the patients who experienced outcomes in each group due to the risk of patient re-identification. We relied on institutional practice of whether cocaine was administered, but we did not have a hard measure of whether patients received cocaine or not. It is possible that some patients may not have received cocaine at a cocaine-using institution, however this is generally the practice when patients have a history or cardiac disease and these patients were excluded in our analysis. It is unlikely the opposite is true, given that there were institutional policies to not use cocaine. Finally, the complexity of the surgery and the severity of the sinonasal disease is not recorded in the databases that we used, therefore, we could not account for it. Previous studies however, have demonstrated no difference in operative field visualization and intraoperative blood loss with cocaine compared to other topical anesthetics [7, 26–28]. It is also possible that patients had minor perioperative complications such as tachycardia or hypertension not reportable as a major cardiac event or perioperative complication.

## Conclusions

Using a large population-based dataset, we found no significant difference in postoperative cardiac related outcomes and death between patients treated with topical cocaine versus that not treated with cocaine. Further, we found a very low rate of cardiac events and mortality in general for patients undergoing these procedures. We have demonstrated with the best available evidence that the rate of cardiac events and death is extremely low in patients undergoing sinonasal surgery regardless of whether they have surgery at an institution using topical cocaine, and that cocaine use does not appear to contribute to peri-operative morbidity or mortality. Further research is needed to determine whether patients may experience minor cardiovascular related complications related to topical cocaine use during sinonasal surgery. Our findings have important implications for physicians conducting sinonasal surgery with a preference for using cocaine as a topical agent during surgery.

## Author Contributions

**Conceptualization:** S. Danielle MacNeil, Brian Rotenberg, Leigh Sowerby.

**Data curation:** Britney Allen, Lucie Richard, Salimah Z. Shariff.

**Formal analysis:** Britney Allen, Lucie Richard, Salimah Z. Shariff.

**Funding acquisition:** Brian Rotenberg.

**Methodology:** S. Danielle MacNeil, Lucie Richard, Salimah Z. Shariff.

**Project administration:** S. Danielle MacNeil, Brian Rotenberg, Leigh Sowerby, Salimah Z. Shariff.

**Supervision:** Brian Rotenberg, Leigh Sowerby, Salimah Z. Shariff.

**Validation:** S. Danielle MacNeil, Britney Allen, Lucie Richard, Salimah Z. Shariff.

**Writing – original draft:** S. Danielle MacNeil, Salimah Z. Shariff.

**Writing – review & editing:** S. Danielle MacNeil, Brian Rotenberg, Leigh Sowerby.

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
