## [Decision Letter · Decision Letter 0]

10 Mar 2020

PONE-D-20-04499

Medical use of cocaine and perioperative morbidity following sinonasal surgery- a population study

PLOS ONE

Dear Dr MacNeil,

Thank you for submitting your manuscript to PLOS ONE. After careful consideration, we feel that it has merit but does not fully meet PLOS ONE’s publication criteria as it currently stands. Therefore, we invite you to submit a revised version of the manuscript that addresses the points raised during the review process.

We would appreciate receiving your revised manuscript by Apr 24 2020 11:59PM. To enhance the reproducibility of your results, we recommend that if applicable you deposit your laboratory protocols in protocols.io, where a protocol can be assigned its own identifier (DOI) such that it can be cited independently in the future. For instructions see: http://journals.plos.org/plosone/s/submission-guidelines#loc-laboratory-protocols

We look forward to receiving your revised manuscript.

Kind regards,

Eng Ooi

Academic Editor

PLOS ONE

Journal Requirements:

2. In ethics statement in the manuscript and in the online submission form, please provide additional information about the patient records used in your retrospective study. Specifically, please ensure that you have discussed whether all data were fully anonymized before you accessed them and/or whether an IRB or ethics committee waived the requirement for informed consent. If patients provided informed written consent to have data from their medical records used in research, please include this information.

3. We noticed you have some minor occurrence(s) of overlapping text with the following previous publication(s), which needs to be addressed:

doi.org/10.1097/MD.0000000000001106

In your revision ensure you cite all your sources (including your own works), and quote or rephrase any duplicated text outside the Methods section. Further consideration is dependent on these concerns being addressed.

Reviewers' comments:

Reviewer's Responses to Questions

**Comments to the Author**

1. Is the manuscript technically sound, and do the data support the conclusions?

Reviewer #1: Yes

Reviewer #2: Yes

Reviewer #3: Yes

2. Has the statistical analysis been performed appropriately and rigorously? 

Reviewer #1: No

Reviewer #2: Yes

Reviewer #3: I Don't Know

3. Have the authors made all data underlying the findings in their manuscript fully available?

Reviewer #1: No

Reviewer #2: Yes

Reviewer #3: Yes

4. Is the manuscript presented in an intelligible fashion and written in standard English?

Reviewer #1: Yes

Reviewer #2: Yes

Reviewer #3: Yes

5. Review Comments to the Author

Reviewer #1: This is a nicely written manuscript on an important topic with a large # of patients across different institutions with opposite practices regarding cocaine use in sinonasal surgery. I generally agree with the conclusions, which do belong in the published literature, but there are some modifications that could make it an even better manuscript.

1. Can the authors comment on the impact, if any, of the differences between groups noted in Table 1? The standardized differences for some reached the threshold of relevance.

2. Some power calculation is needed, especially since the authors are trying to establish a “negative” conclusion.

3. The fisher exact test is not appropriate for large volume data. I believe chi-square is the correct test and do not believe this would change the conclusion. In fact, I suspect the borderline significance of the 30 day findings would be lost (further supporting the authors’ assertions) using the correct but more stringent test.

4. The authors state: “Due to potential risk of patient re-identification, institutional policies prohibit the presentation of results of 5 or fewer individuals. Furthermore, exact p-value for the primary outcome could not be presented to avoid the exact number of event rates from being back calculated.” Although knowing the exact number may not change the conclusions at all, It seems IRB approval could allow identification of the individual cases to determine if there were common features that could be identified to account for the event.

5. Are there any other questions that can be answered from this large dataset – intraoperative events, bleeding, postop pain?

Reviewer #2: Overall an informative study, showing a positive safety profile for cocaine use during sinonasal surgery. This is particularly relevant for those institutions that use cocaine for their procedures. I have suggested the authors consider these points.

Page 4 Line 56-57: Specific reference to Ontario, is this relevant to the paper?

Page 4 Line 61-63: Is there a reference to support the use of cocaine for post operative pain, and additionally the duration of effect.

Page 8 Paragraph starting Line 144: Are the range of cocaine doses known for the institutions that use cocaine?

Could the authors summarize the previous largest studies on cocaine use in sinonasal surgery, there seems to be a lack of discussion of the literature. (major revision)

Page 13 Paragraph starting Line 224: This paragraph is repetitive and should be restricted.

The authors have postulated that a larger study is not possible. This statement assumes the authors have intimate knowledge of all international databases and their capabilities, which I think is speculative.

Additionally, a statement of “best available data” should instead relate back to why this study is successful given the low event rate, which is the size of the population included.

Page 13 Line 227-228: The sentence starting “We included” is repetitive.

Page 14 Line 242: Is case complexity relevant – could the authors refer to case complexity of already reported cocaine associated morbidity. The previously referenced systematic review showed a range of case complexities at risk.

Reviewer #3: The statistical analysis appears sound, but Im not a statistician so I did not delve into the statical analysis of the study. Overall this study will appeal to ENT surgeons who still use cocaine for their surgical patients, thus appeal to a narrow range of readers.

6. PLOS authors have the option to publish the peer review history of their article (what does this mean?). If published, this will include your full peer review and any attached files.

Reviewer #1: No

Reviewer #2: Yes: Dr Jae Murphy

Reviewer #3: No

---

## [Author Response · Author response to Decision Letter 0]

16 May 2020

Thank-you for the comments and reviews. All of the reviewers and editors comments have been addressed in the "response to reviewers" file. We have made the necessary changes to the manuscript.

---

## [Editor Report · Decision Letter 1]

25 May 2020

PONE-D-20-04499R1

Medical use of cocaine and perioperative morbidity following sinonasal surgery- A population study

PLOS ONE

Dear Dr. MacNeil,

Thank you for submitting your manuscript to PLOS ONE. After careful consideration, we feel that it has merit but does not fully meet PLOS ONE’s publication criteria as it currently stands. Therefore, we invite you to submit a revised version of the manuscript that addresses the points raised during the review process.

We look forward to receiving your revised manuscript.

Kind regards,

Eng Ooi

Academic Editor

PLOS ONE

Additional Editor Comments (if provided):

The responses and revisions with regards to addressing the reviewers comments have been useful. I have a comment and recommend a minor revision. Line 66 to 67 ... lasts for hours after the patient is reversed from anesthesia.... references indicate that a second application of cocaine is needed within an hour of the first application to continue significant effects. Can you please specify how long, instead of simply stating for hours, you would expect the anesthetic effect to last for from the initial application of intranasal cocaine in preparation for endoscopic sinus surgery.

---

## [Author Response · Author response to Decision Letter 1]

30 Jun 2020

The reviewer had concern about one of the sentences in the introduction. This sentence has been deleted.

---

## [Editor Report · Decision Letter 2]

7 Jul 2020

Medical use of cocaine and perioperative morbidity following sinonasal surgery- A population study

PONE-D-20-04499R2

Dear Dr. MacNeil,

We’re pleased to inform you that your manuscript has been judged scientifically suitable for publication and will be formally accepted for publication once it meets all outstanding technical requirements.

Kind regards,

Eng Ooi

Academic Editor

PLOS ONE
---

## [Editor Report · Acceptance letter]

20 Jul 2020

PONE-D-20-04499R2 

Medical use of cocaine and perioperative morbidity following sinonasal surgery- A population study 

Dear Dr. MacNeil:

I'm pleased to inform you that your manuscript has been deemed suitable for publication in PLOS ONE. Congratulations! Your manuscript is now with our production department. 

Kind regards, 

on behalf of

Associate Professor Eng Ooi 

Academic Editor

PLOS ONE